# Boosting the Humidity Performances of Na_0.5_Bi_x_TiO_3_ by Tuning Bi Content

**DOI:** 10.3390/nano12142498

**Published:** 2022-07-21

**Authors:** Xiaoqi Xuan, Li Li, Tiantian Li, Jingsong Wang, Yi Yu, Chunchang Wang

**Affiliations:** Laboratory of Dielectric Functional Materials, School of Materials Science & Engineering, Anhui University, Hefei 230601, China; b20201065@stu.ahu.edu.cn (X.X.); b20101003@stu.ahu.edu.cn (L.L.); b51914048@stu.ahu.edu.cn (T.L.); b20301123@stu.ahu.edu.cn (J.W.)

**Keywords:** hydrothermal method, sensors, humidity sensing, linear response

## Abstract

In the field of humidity sensors, a major challenge is how to improve the sensing performance of existing materials. Based on our previous work on Na_0.5_Bi_0.5_TiO_3_, a facile strategy of tuning the Bi content in the material was proposed to improve its sensing performance. Na_0.5_Bi_x_TiO_3_ (x = 0.3, 0.35, 0.4, 0.45) nanocomposites were synthesized by a hydrothermal method. Humidity sensing properties of these nanocomposites were investigated in the relative humidity range of 11% to 95%. Our results show that, compared to the sensor based on nominally pure sample (Na_0.5_Bi_0.5_TiO_3_), the sensor based on Na_0.5_Bi_0.35_TiO_3_ exhibits boosted sensing performance of excellent linear humidity response in the humidity range of 11–75% relative humidity, lower hysteresis value, and faster response/recovery time. The improvement of the sensing performance was argued to be the reason that the proper reduction in Bi content leads to a minimum value of oxygen-vacancy concentrations, thereby weakening the chemical adsorption but enhancing the physical adsorption. These results indicate that the proper underdose of the Bi content in Na_0.5_Bi_0.5_TiO_3_ can greatly boost the sensing performance.

## 1. Introduction

Humidity sensors play an indispensable role in our daily life [1,2]. The monitoring and controlling of environmental humidity is an important subject in meteorology, medicine, and agriculture. Many techniques, such as photon and optical fiber type, fluorescence, surface acoustic wave, field effect transistor, and quartz crystal microbalance have been used to detect the relative humidity (RH) of environment [3,4,5]. Depending on the sensing mechanism, humidity sensors can be mainly clarified into resistance-type and capacitance-type sensors. Resistive sensors show many advantages over capacitive sensors in terms of humidity sensitivity, cost, and mass production [6,7,8]. In recent years, semiconductor nanomaterials have attracted considerable attention as a potentially highly sensitive material. Their high surface volume ratio and nanostructure may increase their interaction with water molecules [9].

In recent years, most researchers focus on improving humidity sensing performance by developing new sensing materials with different nanostructures [10]. Ferroelectric perovskite ceramics have emerged as promising humidity sensing materials due to their large surface volume ratio and exceptional chemical stability [11]. Among them, the lead-free ferroelectric material, Na_0.5_Bi_0.5_TiO_3_, which is often used for energy storage has been evidenced to show excellent humidity sensing properties [12,13,14]. Na_0.5_Bi_0.5_TiO_3_ is considered as a low-cost and environmentally friendly candidate for humidity sensitive materials in humidity detection. Its excellent water adsorption ability can be attributed to its unique characteristics, namely, it has many active sites when chemically reacting with hydrogen and oxygen molecules [14]. Since water molecules are polar, they can be coupled with these active sites forming complex defects for charge compensation. This causes a sharp change in impedance [15]. Our previous work revealed that the humidity sensor based on Na_0.5_Bi_0.5_TiO_3_ nanospheres showed more than five orders of magnitude of sensing response in the relative humidity range of 11–94% [14]. Apart from high sensitivity, a high-performance humidity sensor should have good linearity, small hysteresis loop, and good repeatability. However, the Na_0.5_Bi_0.5_TiO_3_ sensor shows a maximum hysteresis value of 13.5% [14]. Meanwhile, the humidity sensor also exhibits poor linearity, especially in low humidity range. These facts indicate that the performance of the Na_0.5_Bi_0.5_TiO_3_-based sensor still awaits modification.

It has been reported that the A and B ion concentrations and types of Na_0.5_Bi_0.5_TiO_3_ played an important role in the sensing properties [16]. This result motivates our interest to modify the sensing properties of the Na_0.5_Bi_0.5_TiO_3_-based sensors by varying the chemical composition of Bi content. Our results indicate that, when the Bi content is decreased to a certain proportion, the chemical adsorption is weakened due to the decrease of oxygen vacancy concentration. As a result, the physical adsorption becomes clear, which makes the sensor sensitive to humidity in low humidity environments. Therefore, the sensor shows excellent linearity in the humidity range of 11–75% RH. In addition, the humidity sensor has better response speed and less hysteresis than those of the pure Na_0.5_Bi_0.5_TiO_3_—based sensor.

## 2. Experimental Section

### 2.1. Material Preparation

Na_0.5_Bi_x_TiO_3_ (x = 0.3, 0.35, 0.4, 0.45, 0.5) nanomaterials were prepared by the hydrothermal synthesis method using Bi(NO_3_)_3_·5H_2_O, NaOH, and Ti(OC_4_H_9_)_4_ as raw materials. First, 7.2018 g NaOH and 1.455 g Bi(NO_3_)_3_·5H_2_O were respectively added into beakers 1 and 2 containing 15 mL of deionized water to form transparent solutions. Second, 2.04 mL tetrabutyl titanate Ti(OC_4_H_9_)_4_ was dropwise added into the beaker 2 solution to form a mixed solution. The NaOH solution was added into the mixture drop by drop, while stirring for 2 h. Finally, the above mixture was shifted to a 50 mL Teflon-lined autoclave and the reaction was carried out at 180 °C for 24 h. After thoroughly cleaning with deionized water and ethanol, the resultant white powder was dried in a vacuum oven at 80 °C. To prepare samples with different Bi contents, the amount of Bi(NO_3_)_3_·5H_2_O was changed to 0.873, 1.0185, 1.164, and 1.3095 g. The above experiment steps were repeated to achieve Na_0.5_Bi_x_TiO_3_ (x = 0.3, 0.35, 0.4, 0.45).

### 2.2. Material Characterizations

X-ray diffraction was used to study the crystalline phase of the powder (Rigaku Smartlab Beijing Co., Ltd., Beijing, China). Characterization of surface micromorphology was performed on a field emission scanning electron microscopy (FESEM, Regulus 8230, Hitachi Co., Ltd., Tokyo, Japan). The valence states for the ions in samples were measured by X-ray photon spectroscopy (XPS) with Thermo-Fisher ESCALAB 250Xi. The specific surface areas of the samples were tested by Micromeritics ASAP 2460 Brunauer-Emmett-Teller (BET, Shanghai, China) equipment with N_2_ as the carrier gas. 

### 2.3. Sensor Fabrication and Performance Measurements

The impedance-type humidity sensors based on Na_0.5_Bi_x_TiO_3_ powders were fabricated by the aerosol deposition method. Details about the sensor fabrication can be found in our previous paper [14]. In short, 0.05 g nano-powder was first mixed with 10 mL ethanol under ultra sounding to form a homogeneous paste. Then, the paste was sprayed evenly on the Al_2_O_3_ substrate covered with Au interdigitated electrodes. Finally, two copper wires were stuck on the electrodes with silver paste and the as-prepared sensors were dried at 100 °C for 20 min. The impedance of the sensors was measured using a Wayne Kerr 6500B impedance analyzer at room temperature fixed at 25 °C. Six jars containing different types of saturated solutions (LiCl, MgCl_2_, Mg(NO_3_)_2_, NaCl, KCl, and KNO_3_) were used to create different humidity conditions of 11, 33, 54, 75, 85, and 94% RH, respectively. The advantage of using saturated solutions is that the humidity environment is stable and can be changed quickly.

## 3. Results and Discussion

### 3.1. Material Characterizations

Figure 1a shows the XRD patterns of the synthesized Na_0.5_Bi_x_TiO_3_ powders (x = 0.3, 0.35, 0.4, 0.4, 0.45, 0.5). It can be seen from the figure that each sample is a pure perovskite structure without any second phases. The SEM images revealed that the samples are composed of nanospheres. The mean diameter sizes of the nanospheres were found to be 0.270, 0.292, 0.297, 0.286, and 0.336 µm for the samples with x = 0.3, 0.35, 0.4, 0.45 and 0.5, respectively. This result indicates that the reduction in Bi content causes a small change in particle size. The BET surface areas of the samples with x = 0.3, 0.35, 0.4, 0.45 and 0.5 are 11.7171, 9.2850, 5.5804, 5.6924, and 5.3289 m²/g, respectively. The BET surface areas of the samples are very close and almost in the same order of magnitude, which indicates that the BET surface area cannot be a decisive factor that determines the humidity performances. 

### 3.2. Humidity Sensitive Properties

The impedance data of all sensors at 200 Hz during ascending RH run are shown in Figure 2a. The frequency 200 Hz was used since it was evidenced to be the optimal testing frequency [14]. It can be seen that the Bi content has no clear influence on the humidity sensitivity. However, the linearity strongly depends on the Bi content. The best linearity in 11–75% RH is found when x = 0.35. The result of the linear fitting yields the figure of merit R^2^ = 0.999 [Inset], which confirms the outstanding linear humidity response. Therefore, the sample with x = 0.35 is selected as a model sample and detailed characterizations will be performed on this sample. Figure 2b shows the hysteresis behavior curve of the Na_0.5_Bi_0.35_TiO_3_—based sensor. The hysteresis value was computed from the formula [17]: (1)$$\left[\frac{\log\left(Z_{ads}\right)-\log\left(Z_{des}\right)}{\log\left(Z_{ads}\right)}\right]\times 100\%$$
where $Z_{ads}$ is the impedance value for the adsorption process and $Z_{des}$ is the impedance value for the desorption process. The linear and narrow hysteresis loop indicate an easy adsorption and desorption of water vapor during these processes. The maximum hysteresis value of 10% @78% RH is found, which is smaller than 13.5% of the sensor based on pure sample. This finding indicates that the proper underdose of Bi content can improve the hysteresis behavior of the Na_0.5_Bi_0.5_TiO_3_—based humidity sensor.

The repeatability of the Na_0.5_Bi_0.35_TiO_3_ sensor shown in Figure 3a was evaluated by switching the sensor between 11 and 94% relative humidity for 7 cycles with 5 min of dwell time in each RH level. The impedance value hardly changes, and the value under 94% RH remains 94.75% of the initial value after 7 cycles. This fact indicates that the Na_0.5_Bi_0.35_TiO_3_—based sensor exhibits good repeatability. 

The response/recovery curve of the Na_0.5_Bi_0.35_TiO_3_—based sensor is shown in Figure 3b. The measurement of the curve is performed by sequentially placing the sensor in 11–94–11% RH humidity environments (5 min for each humidity environment). The response and recovery time was defined as the time needed for the sensor to achieve 90% of the total impedance variation during the adsorption and desorption process [18,19]. From Figure 3b, the response and recovery times have been found to be 4.4 and 29.5 s, respectively. Both times are superior to 43 and 83 s for the sensor based on the nominally pure sample (Na_0.5_Bi_0.5_TiO_3_) [14]. The results evidence that the underdose of Bi content can notably improve the response/recovery time of the Na_0.5_Bi_0.5_TiO_3_—based humidity sensor.

The long-term stability curves at various humidity levels have been measured on the same equipment with an interval of 30 days. The results were presented in Figure 3c. The impedance values for the selected RH environments are almost constants, indicating that the effect of aging has no significant impact on the performance of the sensor.

The dynamic response of the impedance was recorded by exposing the sensor back and forth between different humidity levels and 11% RH (Appendix A). The recovery/response time deduced between 11% and the remaining RH levels are shown in Appendix A. The humidity properties of Na_0.5_Bi_x_TiO_3_ (x = 0.3, 0.4, 0.45)—based sensor were given in Appendix A.

### 3.3. Mechanism of Sensing Performance Enhancement

To decipher the mechanism of the sensing performance enhancement, the complex impedance plots of Na_0.5_Bi_x_TiO_3_ in different humidity environments were measured. Z-view was used for impedance fitting. It is seen that, at the low RH level of 11%, the complex impedance plots for all samples behave as a semicircle. This is due to the fact that, in the low RH range, water molecules are chemically adsorbed on the surface of the material. Namely, a small number of water molecules interact with charged defects, such as Bi vacancies on the surface of Na_0.5_Bi_x_TiO_3_, resulting in a discontinuous chemisorption layer. Since the electrostatic field of the defects can dissociate the absorbed H_2_O and form OH^−^ groups in the form of [20].
H_2_O → H^+^ + OH^−^(2)

The semicircle in Figure 4a–d represents chemical adsorption. In this case, the electrolytic conduction is difficult. Therefore, the impedance value is large. When the environmental humidity rises, the content of water molecules increases. A continuous hygroscopic layer is formed on the surface and physical adsorption begins to appear on the surface of the material. “A small tail” presenting the typical Warburg impedance [21] appears at the end of the semicircle. The appearance of the Warburg impedance indicates that the physical adsorption is starting to work. As the relative humidity rises to the intermediate range, the number of water molecules attached to the surface increases rapidly. A continuous hygroscopic layer (physically adsorbed water molecules) is formed on the surface of the material. In this layer, the proton (H^+^) is transferred from the hydroxyl group on the surface to the water molecule, forming hydrogen ions (H_3_O^+^) through the ion transfer mechanism of the Grotthuss reaction [22]: H_2_O + H_3_O^+^→H_3_O^+^ + H_2_O(3)

Due to the above mechanism, the impedance curve changes from a high impedance semicircle to a low impedance semicircle as the humidity increases. As a result, the Warburg impedance gradually replaces the semicircle. A careful examination reveals that the earlier the physical adsorption occurs, the better the linear response of humidity sensitivity. For example, for the component x = 0.35, physical adsorption occurs when the relative humidity is 33%, while for other components with poor linearity, physical adsorption does not occur until the relative humidity is 54%. Therefore, there is a close relationship between the linearity of the humidity response and the physical adsorption. This is due to the fact that the early appearance of the physical adsorption leads to a significant drop in impedance in the low humidity range. Consequently, a linear, rather than an upper bending curve of impedance value as a function of relative humidity can be obtained as shown in Figure 2a.

There is a pertinent question: Why does the difference of Bi content affect the appearance of physical adsorption or chemical adsorption? Since the chemical adsorption is related to charged defects, we, therefore, performed XPS measurements. XPS spectra of O 1 s for Na_0.5_Bi_0.35_TiO_3_ and Na_0.5_Bi_0.5_TiO_3_ are displayed in Figure 5 for comparison. In Appendix A, the XPS spectra of the samples with other compositions are described. The O 1 s spectrum of Na_0.5_Bi_0.35_TiO_3_ in Figure 5a can be divided into three Gaussian peaks from low to high binding energy. These peaks located at 532.65, 530.99, and 529.75 eV correspond to oxygen atoms at lattice positions (OL), oxygen vacancies (OM), and chemisorbed oxygen (OH), respectively [23,24]. XPS spectra of Na_0.5_Bi_0.5_TiO_3_ and other samples also show similar XPS spectra to those of Na_0.5_Bi_0.35_TiO_3_. From the XPS spectra, the fractions of oxygen vacancies can be deduced and are listed in Table 1. In this case, the data of humidity response linearity (R^2^) and humidity of physical adsorption (RH_p_) occurrence were also given. One can clearly see that two main factors for the excellent linear humidity response include: (1) Lower RH level for the physical adsorption and (2) lower fraction values of the oxygen vacancies. The former factor leads to a relative rapid decrease in impedance in low RH range as discussed above. For the latter factor, the defects of oxygen vacancies serve as active sites for chemical adsorption. Low fraction value of the defects is unfavorable for chemical adsorption. It is well-known that the humidity sensitivity is the result of chemical adsorption and physical adsorption. When chemisorption is weakened, the sensing material will shield its ferroelectric polarization by physical adsorption. Therefore, the physical adsorption occurs in advance in the lower humidity range.

## 4. Conclusions

In summary, we have investigated the humidity-sensing properties of the impedance-type humidity sensors based on Na_0.5_Bi_x_TiO_3_ (x = 0.3, 0.35, 0.4, 0.45) nanocomposites synthesized by a hydrothermal method. The results reveal that a proper underdose of the Bi content in Na_0.5_Bi_0.5_TiO_3_ can greatly boost the sensing performance. The best humidity performances of high sensing response (over 10^5^), excellent linear response in range of 11–75% RH (R^2^ = 0.999), lower hysteresis value (<10%), faster response/recovery time (4.4/29.5 s) were achieved in the sample of Na_0.5_Bi_0.35_TiO_3_. The performance improvement was believed to be the reason that tuning the Bi content in Na_0.5_Bi_0.5_TiO_3_ can minimize the concentration of oxygen vacancies. Finally, chemical adsorption and physical adsorption can be regulated to achieve the purpose of regulating humidity sensing performance. This work underscores that composition controlling is an effective way to tune humidity sensitivity.

## Figures and Tables

**Figure 1 nanomaterials-12-02498-f001:**
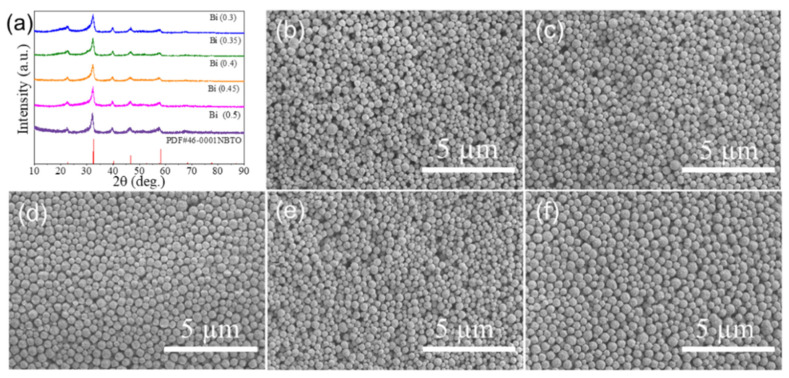
(**a**) XRD patterns and (**b**–**f**) SEM images of the Na_0.5_BixTiO_3_ powders.

**Figure 2 nanomaterials-12-02498-f002:**
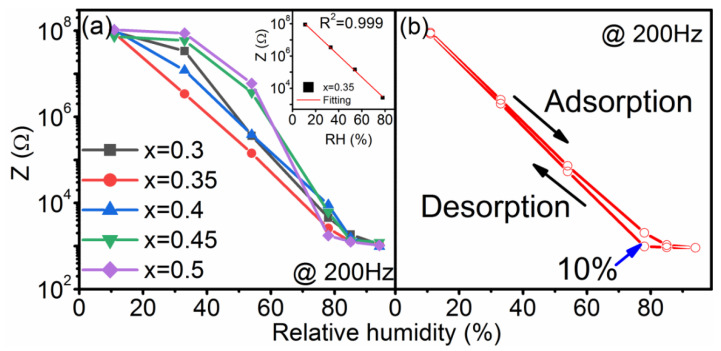
(**a**) Humidity sensing properties of Na_0.5_Bi_x_TiO_3_-based sensor recorded with 200 Hz. The inset graph shows the linear fitting of the impedance data over the humidity range of 11–75% RH for the Na_0.5_Bi_0.35_TiO_3_-based sensor. (**b**) Hysteresis behavior of the Na_0.5_Bi_0.35_TiO_3_—based sensor.

**Figure 3 nanomaterials-12-02498-f003:**
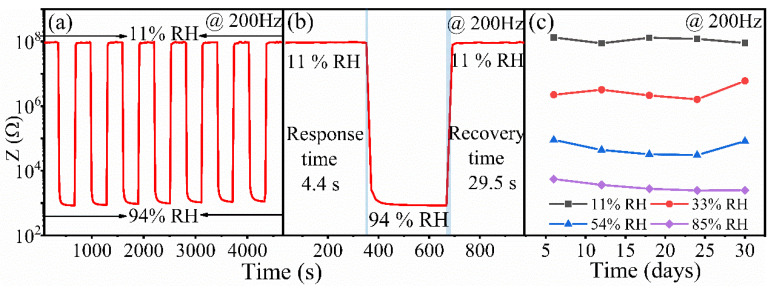
Humidity sensing properties of Na_0.5_Bi_0.35_TiO_3_-based sensor recorded with 200 Hz. (**a**) Repeatability curve, (**b**) response and recovery curve, and (**c**) long-term stability curves under different humidity levels.

**Figure 4 nanomaterials-12-02498-f004:**
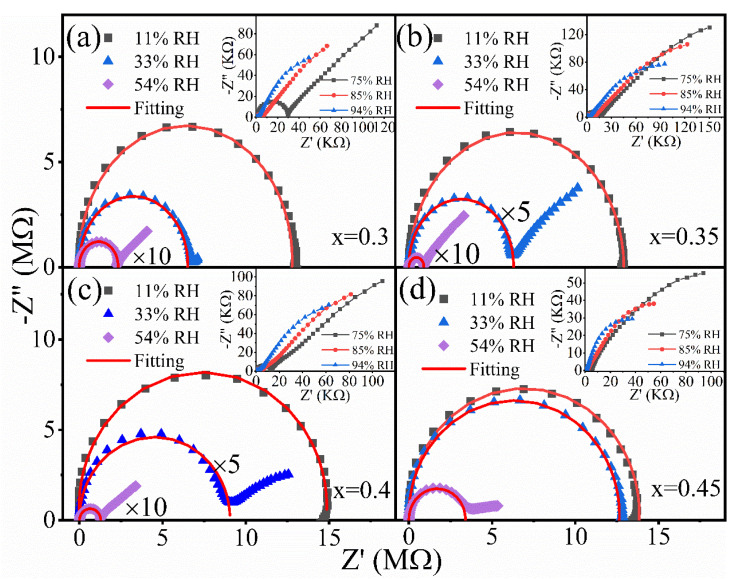
(**a**–**d**) The complex impedance diagrams of Na_0.5_Bi_x_TiO_3_.

**Figure 5 nanomaterials-12-02498-f005:**
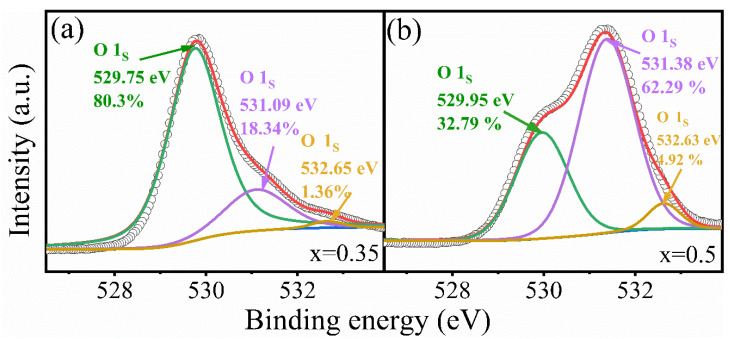
High-resolution XPS spectra of O 1 s of the Na_0.5_Bi_0.35_TiO_3_ (**a**) and Na_0.5_Bi_0.5_TiO_3_ (**b**).

**Table 1 nanomaterials-12-02498-t001:** Humidity response linearity (R^2^), humidity of physical adsorption (RH_p_), and the fractions of oxygen vacancies (f_O_) for the Na_0.5_Bi_x_TiO_3_ samples.

x	R^2^	RH_p_	f_O_ (%)
0.30	0.9484	54%	74.05%
0.35	0.9991	33%	19.25%
0.40	0.9891	33%	29.34%
0.45	0.8484	54%	55.59%
0.50	0.8147	54%	62.29%

## Data Availability

The data presented in this study are available on request from the corresponding author.

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
