# Peer review of "Boosting the Humidity Performances of Na0.5BixTiO3 by Tuning Bi Content"

_nanomaterials, 2022, doi:10.3390/nano12142498_

Round 1

Reviewer 1 Report

Comments are attached.

Reviewer 2 Report

Several compositions of Na0.5Bix-TiO3 (x = 0.3, 0.35, 0.4, 0.45) system were synthesized by a hydrothermal method as the probable candidates for humidity sensor. The impedance type humidity sensors were prepared by the aerosol deposition method. Humidity sensing properties of these materials were investigated in the relative humidity range of 11% to 95%. The sensor based on Na0.5Bi0.35TiO3 shows best sensing performance, namely, excellent linear humidity response in the humidity range of 11–75% RH (R2=0.999), lower hysteresis value (< 10%), and faster response/recovery time (4.4/29.5 s). Therefore, this material has much better sensing characteristics as compared to discussed before Na0.5Bi0.35TiO3 nanocomposite. The authors argued the improvement of the sensing performance by the fact that the proper reducing Bi content can tune the concentrations of Ti3+ ions and oxygen vacancies to minimum values. As a result, composition controlling is an effective way to tune humidity sensitivity. Although the goal of the paper is pure applicative, the result obtained could be useful. The paper can be accepted after minor corrections be introduced.

1.      It would be worthy to discuss the point: has the nanospheres diameter a valuable impact on the sensing characteristics or not? It is seen from Figure 1, the nanospheres diameter is different for different compositions. At least, the mean diameter for every composition should be indicated.

2.      The paper shows an excellent linear fit over a humidity scale of 11-75%, but this  statement is confirmed only by four measured concentrations: 11, 33, 54, 75% RH, and two of them are the boundary points of the humidity range and only two of them are inside. It is reasonable to confirm the linear fit for x=0.35 by 2 or 3 additional concentrations.

3.      The text of the manuscript has to be carefully edited. Several misprints should be corrected in Section 2.1. The phrase “Characterization of surface micromorphology by field emission scanning electron microscopy (FESEM, Regulus 8230, Hitachi Co, Tokyo, Japan)” has no verb. The way of quotation has to be made according to the rules of the journal. 
